# Cholinergic Polarization of Human Macrophages

**DOI:** 10.3390/ijms242115732

**Published:** 2023-10-29

**Authors:** Natalia Roa-Vidal, Adriana S. Rodríguez-Aponte, José A. Lasalde-Dominicci, Coral M. Capó-Vélez, Manuel Delgado-Vélez

**Affiliations:** 1Medical Sciences Campus, University of Puerto Rico, San Juan, PR 00936, USA; natalia.roa@upr.edu; 2Department of Biology, Rio Piedras Campus, University of Puerto Rico, San Juan, PR 00931, USA; adriana.rodriguez51@upr.edu (A.S.R.-A.); corymarie28@gmail.com (C.M.C.-V.); 3Molecular Sciences Research Center, Clinical Bioreagent Center, University of Puerto Rico, San Juan, PR 00926, USA; 4Department of Chemistry, Rio Piedras Campus, University of Puerto Rico, San Juan, PR 00931, USA; 5Institute of Neurobiology, Medical Science Campus, University of Puerto Rico, San Juan, PR 00901, USA; 6Department of Pharmaceutical Sciences, School of Pharmacy, University of Puerto Rico, San Juan, PR 00936, USA

**Keywords:** macrophage, inflammation, acetylcholine receptor, cholinergic receptor, polarization, cholinergic agonist, alpha 7 nicotinic acetylcholine receptor, *CHRFAM7A*, cholinergic, cholinergic anti-inflammatory response

## Abstract

Macrophages serve as vital defenders, protecting the body by exhibiting remarkable cellular adaptability in response to invading pathogens and various stimuli. These cells express nicotinic acetylcholine receptors, with the α7-nAChR being extensively studied due to its involvement in activating the cholinergic anti-inflammatory pathway. Activation of this pathway plays a crucial role in suppressing macrophages’ production of proinflammatory cytokines, thus mitigating excessive inflammation and maintaining host homeostasis. Macrophage polarization, which occurs in response to specific pathogens or insults, is a process that has received limited attention concerning the activation of the cholinergic anti-inflammatory pathway and the contributions of the α7-nAChR in this context. This review aims to present evidence highlighting how the cholinergic constituents in macrophages, led by the α7-nAChR, facilitate the polarization of macrophages towards anti-inflammatory phenotypes. Additionally, we explore the influence of viral infections on macrophage inflammatory phenotypes, taking into account cholinergic mechanisms. We also review the current understanding of macrophage polarization in response to these infections. Finally, we provide insights into the relatively unexplored partial duplication of the α7-nAChR, known as dup α7, which is emerging as a significant factor in macrophage polarization and inflammation scenarios.

## 1. Introduction

Macrophages are a vital component of the immune system responsible for defending the body against pathogens and foreign substances. Found throughout the body, particularly in tissues and organs, macrophages play a crucial role in innate immunity. Their primary function is to engulf and eliminate harmful microorganisms, debris, and dead cells through phagocytic processes involving cytokines [1,2,3,4,5].

Macrophages exhibit remarkable plasticity, capable of adopting different phenotypes and functions in response to signals from their microenvironment [6]. In the presence of pathogens, insults, or tissue alterations, macrophages fulfill their critical role by polarizing and differentiating into distinct subtypes with specific roles and functions. Currently, four main subgroups (M1–M4) have been identified, working in conjunction with other immune cells to defend the body and maintain organ and systemic homeostasis [7]. Importantly, macrophages switch between different functional phenotypes in response to the local cytokine milieu, a process of particular significance during infectious processes [8]. Of note, macrophages are equipped with cholinergic constituents that enable them to respond in an anti-inflammatory manner through a neuroimmune circuit known as the cholinergic anti-inflammatory response (CAR) [9,10]. In macrophages, this route is dependent on the expression of the alpha7 nicotinic acetylcholine receptor (α7-nAChR) [9], an ion channel with anti-inflammatory properties upon stimulation.

Although significant progress has been made in understanding the role and function of the CAR, there are still many aspects that require further clarification and investigation. For instance, the role of macrophage polarization in limiting inflammation through the CAR is still not fully understood. Moreover, the response of polarized macrophages (M1-M4) to the CAR during disease and infection remains understudied. In this review, we aim to discuss the available evidence on how the cholinergic constituents in macrophages contribute to the polarization of these cells towards stimulus-dependent phenotypes. Furthermore, we explore scenarios in which viruses disrupt macrophage polarization and alter α7-nAChR expression, thereby contributing to inflammation. Last, we delve into published research that establishes the connection between the CAR and macrophage polarization, with a particular emphasis on the role of the α7-nAChR and its partial duplication, dup α7.

## 2. Activation and Polarization of Human Macrophages

Human macrophages are white immune cells responsible for destroying pathogens. They also play crucial roles in tissue remodeling, wound healing, angiogenesis, the removal of dead immune cells, metabolism, and the secretion of cytokines and chemokines to stimulate other immune system cells, contributing to the immune response, among other functions. Macrophages become activated and polarized as part of the natural response process that maximizes their antimicrobial properties and maintains host homeostasis. Their cellular plasticity allows them to polarize into two distinct populations: classically activated inflammatory (M1) and alternatively activated anti-inflammatory (M2) macrophages (Figure 1). The latter can be divided into four subsets (M2a, b, c, and d) based on the type of stimulus they receive (Figure 2). The specialization and tropism of macrophages toward organs is an example of cellular plasticity in both health and disease. This is the case with the relatively recent discovery of five types of macrophages (M4, Mhem, M(Hb), HA-mac, Mox, and M17) associated with the development of atherosclerosis and atheroinflammation [11,12,13,14,15,16] (Figure 3). There are also M3 macrophages known as commuting macrophages. The first investigation to look into the origins of this previously unidentified lineage discovered that it was dependent on the switching response between M1 and M2 macrophage pathways [7,17]. It has recently been demonstrated that these macrophages have antitumor activity and limiting tumor proliferation [18], and that they may be a pharmacological target for inducing antitumor immunity [19]. In the context of cancer, there are tumor-associated macrophages (TAMs), and their presence has been linked to metastasis [20]. The polarization state of TAMs is influenced by the tumor microenvironment, especially by cytokines [21]. These TAM-type cells can exhibit phenotypes akin to M1 macrophages [22] and M2 macrophages [23]. Macrophages exhibiting the M17 phenotype can emerge through activation with corticosteroids, granulocyte-macrophage colony-stimulating factor, or IL-10 [7,24]. On the other hand, stimulation with IL-17 imparts an anti-apoptotic signature [7,24]. Finally, M4 macrophages are a subset of macrophages that are polarized by platelet factor 4. This population is found in human lesions and is distinguished by high levels of matrix metalloprotease 7 and S100A8 expression. M4 macrophages are considered atherogenic because they produce proinflammatory cytokines (IL-6 and TNF-α) and have poor phagocytic properties [25,26]. It has been reported that these cells are reproducibly found in coronary artery plaques [27]. In HIV infection, these inflammatory M4 macrophages form a major tissue reservoir of replication-competent HIV-1, which reactivates viral production upon autocrine/paracrine S100A8-mediated glycolytic stimulation [28]. Furthermore, it has been discovered that the level of cholesterol oxidation influences the profile of these M4 macrophages, with LDL being the molecule that causes the most significant changes. This results in the polarization of M2 macrophages and Kupffer cells towards the M4 phenotype. These cells also demonstrate increased neutrophil recruitment and more effective induction of neutrophil extracellular traps [29].

### Polarization of Macrophages during Viral Infection

One of the hallmark events of macrophages during viral infections is their polarization switch during the immune response. This polarization could be triggered by TLR4 or IL-1R ligand activation, IFN-γ binding to its receptor (IFN-γR), interaction of Notch proteins with Delta-like and Jagged ligands, and IL-4 or IL-13 binding to its corresponding receptor [30]. Depending on the virus, the stage of infection, and even the infected person’s gender, macrophages adopt different inflammatory phenotypes: either M1, M2, or a biphasic identity beginning with M1 during the acute phase of infection and then changing to M2 during the chronic phase. Depending on the virus, macrophage polarization plays a variety of roles during viral infection. For example, HIV-1-induced polarization has been shown to influence macrophage susceptibility to infection and replication [31]. The polarization involves the conversion of macrophages into M1 and M2a types through a cytokine-dependent mechanism. Also, studies on the Epstein–Barr virus have shown that M1 macrophage polarization persists even in asymptomatic patients, despite the presence of anti-inflammatory cytokines like IL-10 and TGF-β [32]. Moreover, Japanese encephalitis virus and dengue virus can cause microglia (macrophages from CNS) and infiltrated macrophages to undergo M1 polarization and related proinflammatory activation in mice, and it has been proposed that targeting the occurrence of type 1 immunity may alleviate the pathologically lethal effect of viral encephalitis [33]. Furthermore, during COVID-19, both classically polarized macrophages (M1) and alternatively polarized macrophages (M2) inhibit SARS-CoV-2 infection. However, upon viral infection, M1 and non-activated (M0) macrophages, but not M2 macrophages, significantly up-regulate inflammatory factors [34]. This up-regulation of inflammatory factors is undoubtedly a significant contributor to the inflammation observed in COVID-19 [35]. Additionally, recent studies on the street rabies virus (RABV) have demonstrated its capability to affect macrophage polarization, shifting the macrophages toward an M2-c phenotype. Remarkably, the authors also discovered that a RABV glycoprotein can activate the α7-nAChR in monocyte-derived macrophages (MDMs), thus triggering the cholinergic anti-inflammatory response (CAR) [36]. These collective findings suggest that the RABV can induce an anti-inflammatory phenotype in human macrophages, potentially impacting the functioning of T cells [36]. For comprehensive information on macrophage polarization during viral infections, particularly in noncholinergic settings, we highly recommend consulting the insightful reviews authored by Yu et al. [30] and Atmeh et al. [37]. Likewise, for information concerning macrophage polarization processes during bacterial, fungal, and parasitic infections, please refer to the reviews by Benoit et al. [38], Xie et al. [39], and Zhang et al. [40], respectively.

It is not always the virus itself that polarizes macrophages, but rather viral proteins or cytokines released during infection. This is the case with HIV, where, in addition to the polarizing activity of granulocyte-macrophage colony-stimulating factor (M1) and macrophage colony-stimulating factor (M2) [41,42], the viral protein, Nef, polarizes macrophages to an M1-like phenotype [43]. Similarly, hepatitis C virus core protein engagement with Toll-like receptor 2 of macrophages inhibits M2a, M2b, and M2c macrophage polarization [44]. Furthermore, in the case of a respiratory syncytial virus (RSV) infection, the polarization of alveolar macrophages occurs as a result of cytokines (IFN and GM-CSF), intercellular communication via the Notch–Jagged pathway, and RSV’s direct activation signal [45]. Interestingly, the soluble spike protein of SARS-CoV-2 has recently gained prominence as a potential culprit in the deregulation of macrophage polarization via the α7-nAChR in COVID-19 [46]. The latter is crucial because the activation of this cholinergic ion channel in macrophages plays a significant role in reducing the production of proinflammatory cytokines in these cells [9]. The well-known CAR is responsible for achieving this effect, emphasizing the significance of this nAChR in suppressing inflammation during disease states.

## 3. The Expression of nAChRs and the Potential Role of α7-nAChR in Shaping Macrophage Polarization through the Cholinergic Anti-Inflammatory Response

### 3.1. Expression of nAChRs in Macrophages

Although the presence of neuronal nAChRs in non-neuronal cells, such as macrophages, has long been recognized, we are only now beginning to gain a better understanding of their precise roles and functions. Indeed, we know very little about how these receptors function in macrophages in both health and disease. Macrophages express α1, α2, α3, α4, α5, α6, α7, α9, α10, β1, β2, β3, and β4 subunits [47], resulting in the assembly of various stoichiometries. The expression of nAChRs in macrophages has been extensively studied in humans using primary cultures and representative cell lines from various body tissues. Likewise, macrophages have been extensively examined in a wide range of tissues and cell lines from rats and mice. In recent years, there has been increasing understanding of the roles and functions of nAChRs in macrophages [48,49]. The α7-nAChR is by far the most studied nAChR in human macrophages, as it is known that α7-nAChR-expressing macrophages are effectors of the CAR, with activation resulting in CAR activation and decreased cytokine production [9], ultimately mitigating inflammation.

In human MDMs, the ion-conducting action of the α7-nAChR appears to be regulated by intracellular proteins that restrict its ion-translocation activity [50]. This suggests that the physiological and natural activation of the α7-nAChR depends on intracellular signaling pathways, rather than solely relying on ion conduction through its function as an ion channel. In line with this, recent studies suggest that the α7-nAChR can engage in G protein-coupled receptor-like signaling [51,52,53,54], independent of ion transport through it. In the context of macrophages, this area remains relatively unexplored. For instance, it merits deeper investigation of whether the α7-nAChR is capable of ion transport at the physiological concentrations of endogenous agonists (such as ACh and choline) naturally present in our body or during disease state. Also, it is worth investigating whether exogenous α7-nAChR agonists, such as nicotine, can activate this receptor at concentration levels commonly observed in cigarette smokers. This investigation would provide valuable insights into the potential modulation of α7-nAChR activity by endogenous and external agents and their impact on macrophage polarization and inflammatory response. Surprisingly, smoking is protective in ulcerative colitis, possibly owing to nicotine’s activating α7-nAChRs, causing polarization into M2 macrophages, upregulation of IL-10, and downregulation of IL-6 and TNF-α [55]. Remarkably, only two studies have successfully recorded electrophysiological currents from α7-nAChRs in human macrophages [50,56]. This emphasizes the complexity associated with the intracellular regulation of α7-nAChR activity in macrophages and its response to agonists.

Macrophages are known for their significant cytokine production capacity, and they also possess activatable cholinergic constituents capable of mitigating or inhibiting cytokine production through the CAR. This innate immune response relies on the expression of the α7-nAChR in macrophages. This receptor is physiologically activated by ACh and choline, which are endogenous agonists synthesized by other immune cells [57,58] or by macrophages themselves [47,59]. The efferent vagus nerve can also release ACh to activate the α7-nAChR. The net effect of α7-nAChR activation is that it prevents macrophages from overproducing cytokines [9], allowing inflammation to be controlled through the CAR. However, bacterial [60] and viral infections, such as HIV [61,62], seem to disrupt this delicate innate balance. Therefore, it is not surprising that inflammation becomes a prominent feature in the central axis of bacterial and viral infections.

The finding that viral proteins can bind to and activate the α7-nAChR is highly significant as it reveals the complexity of viral infections in relation to inflammation. It also sheds light on a relatively unexplored area where only a few authors have delved into studying the potential interactions between viral proteins and the anti-inflammatory α7-nAChR. Indeed, viruses such as SARS-CoV-2 [63,64] and RABV [36] have been among the few viruses investigated in this context. It appears that the activation of the α7-nAChR is not universally beneficial, as its activation in HIV-1-infected CD4^+^ T cells promotes HIV-1 transcription [65]. This observation highlights the complex and context-dependent nature of α7-nAChR signaling in different cellular and biochemical environments.

### 3.2. Cholinergic Constituents of Macrophages

ACh has a number of effects on immune cells, with the macrophage playing a key role in dampening inflammation via the CAR. The most common components of the cholinergic system present in immune cells are choline acetyl-transferase (ChAT), vesicular ACh transporters (VAChT), acetylcholinesterase (AChE), and nAChRs. ChAT and VAChT expression has been demonstrated in human alveolar macrophages, lung macrophages, monocytes, and monocyte-derived macrophages (MDMs) [66,67,68], but there is still work to be done to establish the presence of ChAT in macrophages that reside in other tissues and their ability to synthesize/secrete ACh. However, the co-expression of ChAT and VAChT strongly suggests that macrophages have the potential to synthesize and secrete ACh in the lungs and bloodstream. Still in question is whether ACh is stored in macrophages or produced and immediately released without any storage. Experiments in macrophage cell lines have revealed that ACh can operate on nAChRs in a paracrine and autocrine manner to inhibit cytokine production. [47]. AChE enters the picture when it comes time to hydrolyze ACh and end its activity. AChE has been found in human immune cells such as mononuclear lymphocytes and T cells [69], and in human macrophages [70]. Interestingly, α7-nAChR has been shown to interact with AChE in mouse macrophages, which appears to contribute to the reduction of the inflammatory response in macrophages, implicating this enzyme in active anti-inflammatory processes [71]. The cholinergic activation initiated by ACh or choline has the ability to influence the macrophage phenotype. Specifically, it can transform an inflammatory macrophage (M1) into an anti-inflammatory one (M2) by activating the CAR. As a result, it is plausible to consider that this phenotypic conversion, aimed at counteracting excessive inflammation, is also associated with macrophage polarization and subsequent differentiation into the various subpopulations (M3, M4, Mhem, M(Hb), HA-mac, and Mox) that have been identified thus far (see Figure 2 and Figure 3).

### 3.3. The Cholinergic Anti-Inflammatory Response (CAR) Is Closely Associated with the Polarization of Macrophages

In recent years, there has been a growing understanding of the changes in macrophage phenotype following CAR activation. In vivo studies conducted in rats have demonstrated that vagus nerve stimulation, resulting in the release of ACh, leads to a conversion of lung macrophages from the M1 to the M2 phenotype. This conversion is dependent on CAR activation and involves the participation of STAT3 [72]. In murine bone marrow-derived macrophages, it has been shown that the stimulation of α7nAChR activates the STAT3 pro-survival pathway, which serves as a protective mechanism against endoplasmic reticulum stress-induced apoptosis [73]. Additionally, studies conducted on rats with myocardial infarction have demonstrated that the administration of a specific α7-nAChR agonist (PNU 282,987) reduces the infiltration of inflammatory M1 macrophages into the infarcted area and increases the recruitment of M2-type anti-inflammatory peripheral macrophages to the infarcted tissue [74]. Moreover, in vitro studies using rodent cells focused on atherosclerosis have shown that CAR agonists (ACh and GTS-21) suppress M1 macrophage polarization while promoting M2 macrophage polarization through the upregulation of TNFAIP3 and phosphorylation of STAT3 [75]. This response strongly supports the presence of anti-inflammatory macrophages (M2) to counteract atheroinflammation. Importantly, this study provides evidence that CAR plays a role in the development of atherosclerosis by regulating macrophage function and promoting polarization toward anti-inflammatory (M2) phenotypes. Furthermore, another study on atherosclerosis using mouse peritoneal macrophages has reported that macrophage polarization is associated with the expression of nAChRs, specifically with the suppression of α4β2-nAChR expression in M1 macrophages [76]. One of the most compelling pieces of evidence supporting the medullary role of the α7-nAChR is the study on preeclampsia conducted by Han et al. [77]. This study provides strong evidence that women with preeclampsia exhibit downregulated levels of α7-nAChR in their decidual macrophages, which is accompanied by a decrease in the number of M2 phenotype macrophages and an increase in the number of M1 phenotype macrophages. The findings of this study demonstrate that the transition from the anti-inflammatory (M2) phenotype to the pro-inflammatory (M1) phenotype in macrophages can occur simply by reducing α7-nAChR levels. The polarization of macrophages appears to have broad implications, as they have also been implicated in the anti-inflammatory effects of acupuncture [78] and electroacupuncture [79].

The precise immunological identity (M1–4) of the macrophages involved in the CAR remains under study. This raises the question of whether M2 macrophages are more avid to activation by ACh or choline compared to M1 macrophages. Given the vast diversity of macrophage subpopulations upon polarization, it is highly improbable that they all exhibit the same response to endogenous ACh and choline. In fact, both published and unpublished findings from our group agree as to the heterogeneity of their responses. For example, electrophysiological studies have demonstrated that only a fraction of MDMs exhibit electrophysiological currents indicative of the α7-nAChR. It is worth noting that the prevalence of these currents varies across different electrophysiological configurations, with approximately 15% observed in outside-out recordings, around 30% in inside-out recordings, and approximately 40% in single-channel current measurements [50], and an even smaller percentage (≈10%) of these cells display calcium mobilization in response to ACh (unpublished results). Recent findings from Victor Tsetlin’s group demonstrated similar results in human macrophages (THP-1). Patch-clamp experiments revealed the presence of currents in approximately 22% of the tested cells [56]. Another crucial consideration is the distinct levels of α7-nAChR expression among different macrophage populations. Could it be that macrophages expressing higher levels of α7-nAChR demonstrate a stronger anti-inflammatory response upon ACh/choline activation? Currently, we lack an answer to this question. However, what we do know is that elevated levels of α7-nAChR induced by the HIV protein, gp120, do not potentiate the anti-inflammatory response (CAR), suggesting a disruption of the CAR [61]. From early stages, these cells already exhibit glimpses of their cellular and functional complexity. We are referring to monocytes, the precursors of macrophages. Studies have shown that primary monocytes can be categorized into two distinct populations based on their nAChR expression levels [61]. Specifically, we have demonstrated that by using primary monocytes from both control subjects and infected individuals [61], and by introducing a selective fluorescent antagonist for α7-nAChR (α-BuTX) followed by analysis using flow cytometry, these cells segregate into two discernible populations. One population displays high levels of α7-nAChR, while the other exhibits low levels. We have observed similar results in peripheral blood mononuclear cells (unpublished results). The phenotype of these monocytes upon differentiation into macrophages remains unknown. Additionally, we have yet to determine whether there are differences in the expression and functionality of the cholinergic constituents, particularly the anti-inflammatory α7-nAChR, among macrophages derived from these two populations. In summary, the precise cellular basis of CAR remains under study. Gaining a better understanding of these processes would offer valuable insights into the pharmacological modulation of inflammation through the α7-nAChR expressed by macrophages.

### 3.4. Experimental Challenges in Testing the Cholinergic Anti-Inflammatory Response in Macrophages: The Choline Issue

There is a significant challenge regarding choline when studying human macrophages in vitro, whether using primary cultures, macrophages or monocytic cancer cell lines. Choline is a precursor and metabolite of ACh. Because endogenous choline serves as an agonist of the α7-nAChR, it is important to know the concentrations present in the culture medium where these cells are grown and to monitor the accumulation of choline during cell culture to avoid desensitization of the α7-nAChR [80]. While macrophages are being cultured, choline can accumulate in the culture medium for various reasons. The first may be due to the release from cellular components. During cell growth and turnover, cellular components containing choline, such as phospholipids, may be broken down and release choline into the cell culture media. This process could contribute to the increased choline levels observed over time. Another possible reason for choline accumulation in the culture medium may be due to choline metabolism by macrophages. Macrophages have enzymatic pathways that metabolize choline, and these pathways may become more active as the cells respond to environmental cues or specific stimuli. The increased metabolic activity could result in higher choline levels as a byproduct or as part of intracellular signaling processes. It is worth mentioning that another important caution is that primary cells such as MDMs and cancer cell lines vary greatly in terms of choline metabolism, whereas cancer cells exhibit altered choline metabolism [81]. Therefore, extrapolation of the results between both types of cells when studying inflammation or cell polarization must be carried out with care. Of note, it was recently discovered that the uptake and metabolism of choline are important for macrophage inflammation because the polarization of primary bone marrow macrophages with lipopolysaccharide (LPS) resulted in an increased rate of choline uptake and higher levels of phosphatidylcholine synthesis. Furthermore, choline uptake and metabolism modulate macrophage IL-1β and IL-18 production [82]. Therefore, could macrophages incorporate choline into themselves for anti-inflammatory purposes? This newly incorporated choline can easily be used by the macrophage to synthesize acetylcholine and release it autocrinely and paracrinely [47].

In the body, the systemic and local concentration of choline varies depending on the organ in which it is measured and the physical status of the person. In healthy adults, the concentration of choline in plasma ranges from 7 to 20 µM [83], whereas that of sick people is usually high [84,85,86] or very low [87]. The desensitizing concentration of choline reported for neuronal α7-nAChR in rats is at concentrations above 10 µM [80,88]. Therefore, if it is desired to study the CAR in MDMs in vitro, care should be taken to keep the choline concentration away from 10 uM to activate the α7-nAChR effectively. It turns out that the accumulation of choline during the cultures of MDMs occurs naturally, even when they are not exposed to any treatment (Figure 4). For example, our studies testing the cholinergic anti-inflammatory response in MDMs (primary culture) exposed to the HIV-1 viral protein, gp120_IIB_, demonstrate a time-dependent accumulation of choline. Said accumulation occurs irrespective of the application of gp120_IIIB_, pyridostigmine, or the co-application of both (Figure 4). Importantly, in these experiments, the culture medium already included choline. After evaluating four different preparations using four different lots of the base (RPMI-1640), fetal bovine serum, and human serum, we identified that the average initial concentration of choline is 113.70 µM ± 0.2. Thus, the concentration of choline exceeds the desensitizing concentration for the α7-nAChR, 10 µM.

Therefore, when it is desired to stimulate the α7-nAChR to activate CAR in macrophages, the previous medium should be removed and the remainder washed and replaced with a fresh, low-choline medium to ensure that the α7-nAChR can be stimulated with the agonist of interest. To activate α7-nAChR, ACh, nicotine, or more elaborate agonists such as GTS-21 and PNU 282,987 are typically used. The application of ACh must be preceded by treatment with pyridostigmine to prevent acetylcholinesterase from hydrolyzing ACh and converting it into choline and acetate, thus contributing to increasing choline concentration at the expense of ACh.

### 3.5. Function and Role of CHRFAM7A in the Cholinergic Anti-inflammatory Response Operation

*CHRFAM7A* is a gene that encodes a protein (called duplicated or dup α7) found in humans, specifically in the brain and immune cells such as macrophages. This fusion gene results from a gene duplication event [89] involving the ULK4 (Unc51-like kinase-4) and the *FAM7A* genes [90]. *CHRFAM7A* has been of interest due to its potential involvement in various physiological processes and diseases. While its exact function is not yet fully understood, studies suggest that *CHRFAM7A* may play a role in modulating cholinergic signaling, including those related to inflammation and cognitive function. According to the available information, *CHRFAM7A* functions as a negative regulator of the expression and activity of the α7-nAChR [91], thereby diminishing the anti-inflammatory activity in macrophages. In macrophages, this suggests that elevated levels of *CHRFAM7A* could potentially lead to an inflammatory phenotype (M1) by inhibiting the activity of the α7-nAChR. In line with this, a recent report by Li et al. identified the expression of *CHRFAM7A* as a promoter of inflammation [92]. Moreover, in a recent study, a transgenic *CHRFAM7A* mouse was employed to examine the impact of this human-specific gene on the development of knee osteoarthritis. The findings revealed that *CHRFAM7A* serves as an exacerbating factor in inflammation and tissue damage characteristic of osteoarthritis [93]. On the contrary, studies conducted with primary macrophages have revealed that co-expression of *CHRNA7* (gene coding for α7-nAChR) and *CHRFAM7A* leads to a greater CAR upon stimulation with lipopolysaccharide (LPS), as compared to macrophages expressing only one of these genes [94]. This indicates a potential modulation or regulation between these two proteins. Similar evidence has also emerged from studies carried out in humanized mouse models of radiation-induced lacrimal gland injury, involving cell types other than macrophages [95]. These studies demonstrate that elevating *CHRFAM7A* levels yields beneficial outcomes in these contexts. These findings emphasize the relevance of *CHRFAM7A* in modulating the inflammatory response mediated by α7-nAChR in vivo and in vitro. This also highlights its significance as a factor to be taken into account during the development and evaluation of drugs targeting α7-nAChR to dampen inflammation.

### 3.6. CHRFAM7A and Macrophage Polarization

The role of *CHRFAM7A* in macrophages extends beyond inflammation and encompasses macrophage polarization as well. Studies have demonstrated that *CHRFAM7A* facilitates M2 macrophage polarization through the Notch pathway, ameliorating hypertrophic scar formation in humans [92]. Furthermore, in a mouse model of renal fibrosis, the overexpression of *CHRFAM7A* was found to suppress M1 macrophage activation following unilateral ureteral obstruction (UUO) when compared to wild-type mice. Conversely, the expression of M2 macrophage markers such as CD206 and FIZZ1 was elevated in mice overexpressing *CHRFAM7A* after UUO, indicating an increased transition of macrophages from an M1 to an M2 phenotype [96]. Nevertheless, in contrast to the findings in the hypertrophic scar mouse model [92], overexpression of *CHRFAM7A* in the central nervous system (CNS) has been shown to mitigate cerebral ischemia–reperfusion injury. This effect is achieved through the inhibition of microglia pyroptosis, which is mediated by the NLRP3/Caspase-1 pathway [97]. Consequently, the upregulation of *CHRFAM7A* expression in macrophages proved to be anti-inflammatory.

Collectively, the existing findings indicate that the impact of *CHRFAM7A* on inflammation is cell dependent. However, comprehensive studies, particularly in human subjects, are limited. Further investigations are required to clarify/elucidate its cellular function and the extent of its involvement in inflammation before considering pharmacological modulation of its activity or expression.

## 4. Conclusions and Future Directions

The information presented here clearly highlights the fundamental role of macrophages in regulating inflammatory processes, with their polarization being associated with the functioning of the CAR, particularly on M1 and M2 phenotypes. However, the role of CAR in the differentiation of M0 macrophages to more sophisticated phenotypes like M3, M4, Mhem, M(Hb), HA-mac, and Mox remains unknown, and the exact order of polarization is yet to be determined. In response to a stimulating signal or insult, the CAR may polarize macrophages, or macrophages may auto-polarize by releasing autocrine polarizing substances, such as ACh [47], which are capable of activating the CAR. The available literature suggests that macrophage polarization depends on the specific insult activating it (virus, bacteria, parasite, fungus, etc.), the macrophage’s location at the time of activation, and the surrounding biochemical microenvironment. Within this chemical environment, endogenous agonists (ACh and choline) released by the body activate α7-nAChR. Studies on influenza [98] and bacterial infections, like *Mycobacterium tuberculosis* [99], have shown increased ACh levels as infections progress, indicating a more complex scenario where ACh release by both host and pathogen may lead to desensitization of α7-nAChR in macrophages and CAR activation. Interestingly, ACh release has also been reported to have bactericidal properties, further influencing host defenses against pathogens [100]. Moreover, recent discoveries, such as ACh functioning as a chemotactic agent for certain bacteria, have opened new scientific directions and added complexity to the intricate mechanisms influencing CAR activity in macrophages [101]. These advancements pave the way for innovative approaches to treat inflammatory problems.

From a therapeutic standpoint, α7-nAChR emerges as a promising option for modulating the CAR in macrophages to counteract local and systemic inflammation. However, optimizing its activation remains a subject of investigation, and understanding the role of *CHRFAM7A* in the pharmacological activation of α7-AChR is critical. It is possible that α7-nAChR may or may not require its inherent ion translocation ability and instead function primarily as a GPCR to exhibit an anti-inflammatory phenotype. Additionally, it is worth considering that the anti-inflammatory profile of macrophages (M2) may not solely rely on the activation of α7-nAChR, as other nAChRs, such as α4β2, also appear to play a contributory role. Therefore, when formulating therapeutic strategies, both α7-nAChR and α4β2 should be taken into account for a comprehensive approach. In conclusion, the future shows significant promise for pharmacologically harnessing the cholinergic constituents expressed by macrophages, and ongoing efforts in the field are moving in that direction.

## Figures and Tables

**Figure 1 ijms-24-15732-f001:**
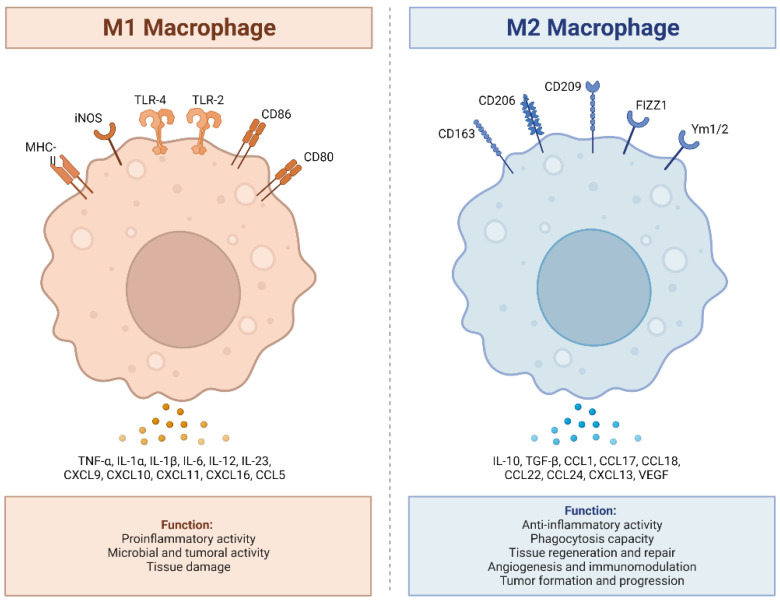
M1 and M2 macrophages exhibit distinct inflammatory phenotypes by expressing and secreting pro-inflammatory and anti-inflammatory cytokines, respectively. M1 macrophages, classically activated, play a vital role in pro-inflammatory responses, expressing receptors like MHC-II, iNOS, TLR-2, TLR-4, CD86, and CD80. They secrete pro-inflammatory cytokines (TNF-α, IL-1α, IL-1β, IL-6, IL-12, IL-23, CXCL9 (MIG), CXCL10, (IP-10), CXCL11 (I-TAC), CXCL16, and CCL5 (RANTES), produce ROS and NO, and promote Th1 responses. In contrast, M2 macrophages, alternatively activated, exhibit anti-inflammatory and tissue repair functions, expressing receptors such as CD163, CD206, CD209, FIZZ1, and Ym1/2. They secrete anti-inflammatory cytokines (IL-10, TGF-β, CCL1 (I-309), CCL17 (TARC), CCL18 (MIP-4), CCL22 (MDC), CCL24 (Eotaxin-2), CXCL13 (BCA-1), and VEGF), enhance phagocytosis, and support Th2 responses. The balance between M1 and M2 states is crucial for immune homeostasis and effective immune responses. Adapted from “Macrophage Polarization: M1 and M2 Subtypes”, by BioRender.com (2019). Retrieved from https://app.biorender.com/biorender-templates (accessed on 23 September 2023).

**Figure 2 ijms-24-15732-f002:**
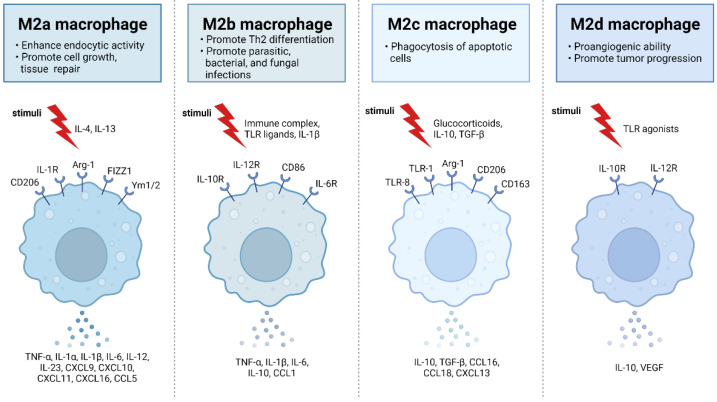
M2 macrophages are divided into M2a, M2b, M2c, and M2d subtypes. These macrophages differ in their cell surface mark ers, secreted cytokines, and biological functions. Once stimulated, each subtype exhibits unique functions and expresses characteristic receptors that influence their distinct roles in immune regulation and tissue homeostasis. M2a macrophages, alternatively activated by IL-4 and IL-13, are involved in tissue repair, high endocytic activity, and anti-inflammatory responses. They express CD206, IL-1R, Arg-1, FIZZ1, and Ym1/2 which contribute to their polarization. M2b macrophages, activated by immune complexes, IL-1β, and Toll-like receptor (TLR) signaling, exhibit regulatory functions in immune responses and Th2 differentiation. Their characteristic receptors include IL-10R, IL-12R, CD86, and IL-6R. M2c macrophages, activated by IL-10, TGF-β, and glucocorticoids, are known for their anti-inflammatory and immune-regulatory functions such as phagocytosis of dying cells. They express TLR-8, TLR-1, Arg-1, CD163, and CD206. M2d macrophages, activated by TLR ligands, are involved in immune modulation and express IL-10R and IL-12R. Adapted from “M2 Macrophage Subtypes”, by BioRender.com (2019). Retrieved from https://app.biorender.com/biorender-templates (accessed on 23 September 2023).

**Figure 3 ijms-24-15732-f003:**
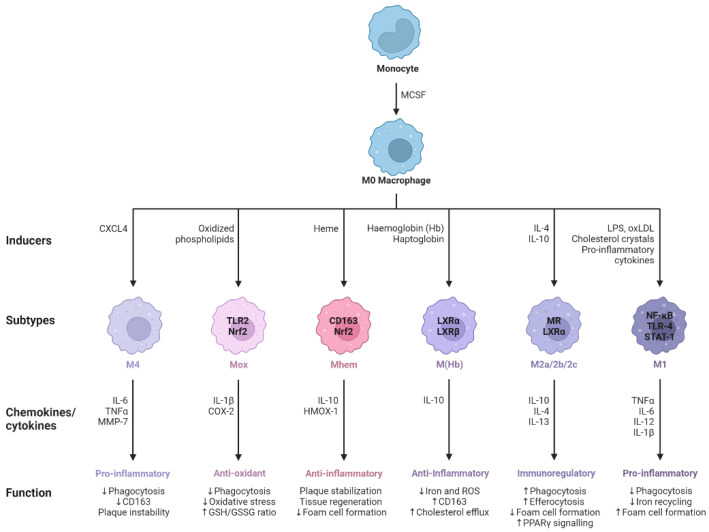
M0 macrophages are the precursors that give rise to specialized subtypes found in atherosclerotic lesions. Monocytes are differentiated to M0 macrophages via monocyte colony stimulating factor (MCSF). Depending on the tissue microenvironment, the M0 macrophage is polarized to a particular subtype. The primary subtypes are pro-inflammatory M1 and anti-inflammatory M2 macrophages with immunoregulatory properties. In addition, multiple other subtypes have been identified including M(Hb), Mhem, Mox, and M4. All of these subtypes produce various chemokines and cytokines and have different functions within the atherosclerotic plaque. Arrows pointing upwards indicate an increase, whereas arrows pointing downwards indicate a decrease. Adapted from “Macrophage Subtypes in Atherosclerosis”, by BioRender.com (2014). Retrieved from https://app.biorender.com/biorender-templates (accessed on 23 September 2023).

**Figure 4 ijms-24-15732-f004:**
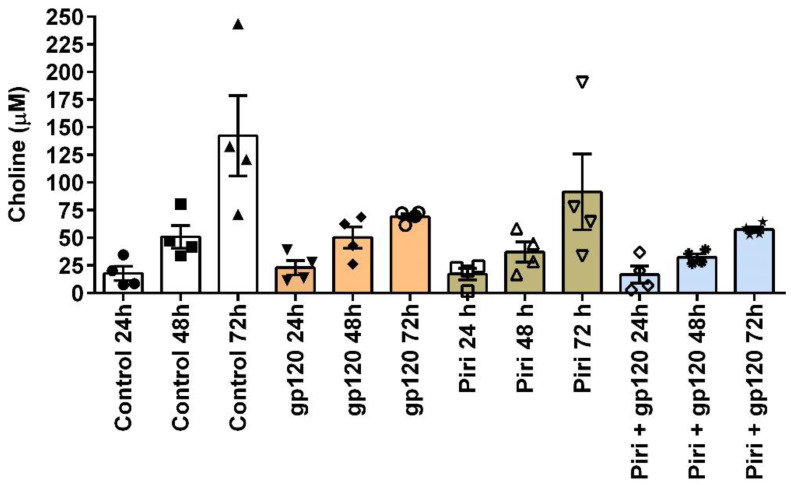
Choline concentrations in primary cultures of MDMs. Choline levels were determined using Biovision’s Choline/Acetylcholine Quantification Colorimetric/Fluorometric Kit (Biovision, Milpitas, CA, USA; cat# K615-100) following the manufacturer’s instructions. The graph was constructed by subtracting the choline values from the supplemented culture medium (initially 113.70 uM) to show the choline concentration produced exclusively by the MDMs. The data were derived from four independent experiments, and error bars represent the standard error of the mean (SEM). Over time, the choline concentration gradually increased for all tested conditions, underscoring the significance of considering choline accumulation when aiming to activate the cholinergic anti-inflammatory response through the α7-nAChR. The control condition consisted of MDMs without any treatment, incubated for 24, 48, and 72 h. A one-way ANOVA followed by a Sidak’s multiple comparisons test revealed no significant difference when comparing all conditions against controls at 24, 48, and 72 h. The cell culture medium used was RPMI-1640 (Sigma, St. Louis, MO, USA) supplemented with 20% inactivated fetal bovine serum (Sigma, St. Louis, MO, USA), 10% inactivated human serum (Sigma, St. Louis, MO, USA), 2 μg/mL of macrophage colony-stimulating factor (Invitrogen, New York, NY, USA), and 1% Penicillin Streptomycin (Sigma, St. Louis, MO, USA). The abbreviation “Piri” refers to pyridostigmine (Sigma, St. Louis, MO, USA); gp120 refers to gp120_IIB_.

## Data Availability

Data is available upon request, subject to privacy restrictions outlined in the IRB protocol.

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
