# Peer review of "Cholinergic Polarization of Human Macrophages"

_ijms, 2023, doi:10.3390/ijms242115732_

Round 1

Reviewer 1 Report

The topic of this manuscript is extremely interesting and novel, unfortunately numerous weaknesses reduce the quality of the manuscript.

First of all the lack of clarity and fluidity of the text which makes reading difficult, and the presence of entire   long paragraphs without references.  Thus, the manuscript requires careful revision and drastic changes.

Some of the main critical issues:

Pag.1 line 44:  The authors devono completare questa  sentenza generale : Early classification of macrophage polarization defined classically activated, pro-inflammatory M1 and alternatively activated, generally anti-inflammatory M2 phenotypes, but, macrophages  can switch between different functional phenotypes according to local cytokine milieu. Thus other subtypes have also been described

Pag. 2 line 74: moreover, tumor-associated macrophages (TAMs), which often have an M2 phenotype and promote tumor progression, are known . Recently, however, some TAMs were found to express both M1 and M2 markers.

Pa.2 line 80 the sentence “appearance  depended on the switching response between M1 and M2 macrophage pathways” is the same of not cited manuscript of “de Sousa JR, Da Costa Vasconcelos PF, Quaresma JAS. Functional aspects, phenotypic heterogeneity, and tissue immune response of macrophages in infectious diseases. Infect Drug Resist. 2019 Aug 22;12:2589-2611. doi: 10.2147/IDR.S208576”.

In  the manuscript  of de Sousa, was reported also that “Lastly, the concept of cell emergence in the case of M17 macrophages is new. One of the earliest studies supporting this hypothesis showed that cell formation can be induced by corticosteroids, granulocyte-macrophage (GM)-CSF, or IL-10” which should be reported for completeness of description

Pag.4 line 130: If the authors describe Polarization of macrophages during viral infection , they must describe also Macrophage Polarization in Bacterial or Parasitic Infections (mentioning Marie Benoit, Benoît Desnues, Jean-Louis Mege; Macrophage Polarization in Bacterial Infections. J Immunol 15 September 2008; 181 (6): 3733–3739. https://doi.org/10.4049/jimmunol.181.6.3733 and Zhang J, Sun Y, Zheng J. The State of Art of Extracellular Traps in Protozoan Infections (Review). Front Immunol. 2021 Dec 14;12:770246. doi: 10.3389/fimmu.2021.770246, for example…).

Pag 5 line 158-167 should be moved into the paragraphs were  α7-nAChR  was described/discussed

Pag.8 line 318  the statement that “primary monocytes can becategorized into two distinct populations based on their nAChR expression levels” needs to be more thoroughly discussed

The authors must use the same units of measurement, at pag 10, theu use micromol/L  at line 360 and  μM  at line 362

The section 3.5 is very confused and disorganized. Often jumping from one concept to another causes continuous acrobatics for readers.

Plagiarism in the  text was detected   by plagiarism detection software

Author Response

Dear Reviewer,

Attached, you will find my responses to your critiques. Thank you for your valuable comments.

Reviewer 2 Report

This review on Cholinergic Polarisation of Human Macrophages is quite interesting, raising several relevant issues and showing evidence for the own lab for some specific characteristics of polarised macrophages. 

I only have minor comments to make, which should be easily addressed.

If possible, rephrase lines 65-67 in order for the sentence to have more cohesion. 

The authors refer mainly as an example macrophage polarisation towards viruses. Would it be possible to include a small paragraph on how different is this mechanism relative to macrophages polarisation in the presence of bacteria, parasite and fungus?

The authors mention the alternative macrophages M4, Them, M(Hb), HA-mac and Max but only describe M4 in the text, leaving some functions of the remaining macrophages in figure 3. Could the authors also briefly describe the function of these macrophages in the text?

In lane 139 the authors mention that HIV-1-induced polarisation influences macrophage susceptibility to infection. Can the authors briefly explain how based on the reference they cite?

In lanes 232 and 233 the authors mention that bacterial and viral infections disrupt the innate balance of pro-inflammatory vs anti-inflammatory nAChR mediated activation. Can you explain how this disruption is achieved?

Can the authors rephrase sentences in lanes 246, 247 and 248?

Although the topic of this review are human macrophages, can the authors briefly introduce mouse models as an alternative in vivo setting (to the in vitro that possesse several limitations) to study macrophage function?

The english quality is overall good. I just mention a few sentences that should be changed in order to become clearer and with a more logic flow. 

Author Response

(The authors gave the same response as above.)

Reviewer 3 Report

I have following comments: 

1.     This study is not clear in many aspects. Please prepare a table of transcriptomic studies of macrophage (MAC) development (if available), either from Single-cell RNA-seq or Bulk RNA-seq. This table with citations will benefit the readers by differentiate MAC through their molecular or genetic characterizations. 

2.     Please prepare a table of transcriptional regulation or miRNA modification of MAC subsets under either infectious diseases or cancer research or etc. This table will present clear goal of your review study, which can be related to clinical implications. 

3.     Please pay attention to your usage of words, such as machinery for cholinergic, which is obviously a signaling pathway to transmit messages into nuclear machinery for further processing. Also, your usage of “CAR”, “CAP” to represent cholinergic anti-inflammatory response (CAR) or pathway is confusing to readers. 

Please check the word usage. 

Author Response

(The authors gave the same response as above.)

Round 2

Reviewer 1 Report

The authors have sufficiently improved the manuscript in accordance with the reviewer's comments

Reviewer 3 Report

N/A